# Optoacoustic Imaging in Inflammation

**DOI:** 10.3390/biomedicines9050483

**Published:** 2021-04-28

**Authors:** Adrian P. Regensburger, Emma Brown, Gerhard Krönke, Maximilian J. Waldner, Ferdinand Knieling

**Affiliations:** 1Department of Pediatrics and Adolescent Medicine, University Hospital Erlangen, Friedrich-Alexander-Universität (FAU) Erlangen-Nürnberg, Loschgestr. 15, D-91054 Erlangen, Germany; adrian.regensburger@uk-erlangen.de; 2Department of Physics, University of Cambridge, JJ Thomson Avenue, Cambridge CB3 0HE, UK; emma.brown@cruk.cam.ac.uk; 3Cancer Research UK Cambridge Institute, University of Cambridge, Li Ka Shing Centre, Robinson Way, Cambridge CB2 0RE, UK; 4Department of Medicine 3, University Hospital Erlangen, Friedrich-Alexander-Universität (FAU) Erlangen-Nürnberg, Ulmenweg 18, D-91054 Erlangen, Germany; gerhard.kroenke@uk-erlangen.de; 5Department of Medicine 1, University Hospital Erlangen, Friedrich-Alexander-Universität (FAU) Erlangen-Nürnberg, Ulmenweg 18, D-91054 Erlangen, Germany; maximilian.waldner@uk-erlangen.de

**Keywords:** optoacoustics, photoacoustics, imaging inflammation, MSOT, RSOM, PAI, acute inflammation, chronic inflammation, molecular imaging

## Abstract

Optoacoustic or photoacoustic imaging (OAI/PAI) is a technology which enables non-invasive visualization of laser-illuminated tissue by the detection of acoustic signals. The combination of “light in” and “sound out” offers unprecedented scalability with a high penetration depth and resolution. The wide range of biomedical applications makes this technology a versatile tool for preclinical and clinical research. Particularly when imaging inflammation, the technology offers advantages over current clinical methods to diagnose, stage, and monitor physiological and pathophysiological processes. This review discusses the clinical perspective of using OAI in the context of imaging inflammation as well as in current and emerging translational applications.

## 1. Introduction

The typical constellation of acute inflammation encompasses four cardinal signs: reddening, local warming, swelling, and pain. On a smaller scale, inflammation presents a reaction to local damage or an infection resulting in a short ischemic phase followed by hyperemia of the local vasculature, remodeling of the connective tissue, and invasion of inflammatory cells [1]. The entire process is well-conserved and can be triggered by multiple stimuli including allergens, antimicrobial pathogens, injury, burns, wounds, or chemical irritants. If the process is not resolved, ongoing inflammation leads to chronic tissue damage and remodeling resulting in diseases such as cancer, arthritis, obesity, diabetes, and neurodegeneration [2,3]. The ability to visualize these processes non-invasively in vivo using molecular imaging modalities has fostered our understanding of the underlying pathophysiology [4] and aids the development of personalized drug regimens [5].

This review discusses the unique ability and scalability of optoacoustic or photoacoustic imaging (OAI/PAI) to visualize, monitor, and understand the molecular mechanisms of inflammation. OAI encompasses a set of heterogenous optical imaging technologies utilizing the photoacoustic effect. Based on the observation of Alexander Graham Bell that there is formation of sound waves following light absorption [6], the principle has been rediscovered and used to develop imaging approaches [7]. In OAI, optoacoustic contrast arises when the light is absorbed by tissue molecules and converted into acoustic pressure waves, which can be recorded and formed into optoacoustic images [8].

Due to their unique absorption characteristics, endogenous molecules such as deoxyhemoglobin, oxyhemoglobin, collagen, lipids, and melanin enable OAI in the near-infrared (NIR) range [8,9] (Figure 1). In the future, targeted exogenous probes or contrast agents may further enrich the clinical capability of this technology by specifically labeling target molecules or cells for more personalized imaging approaches [6,10,11,12,13]. OAI is able to detect a broad range of endogenous and exogenous molecules within a single imaging modality whilst maintaining high spatial resolution at greater depths compared to optical imaging, due to the detection of ultrasound (US) waves, which scatter less in tissue than light [7,14]. The non-invasive nature of OAI will allow broad application to both experimental and clinical settings without interfering with biological processes. This review briefly describes the scalability of OAI and its current and potential ability to image inflammatory conditions from a clinical perspective, with regard to different organ systems as follows: cardiovascular, dermatologic, gastrointestinal, musculoskeletal, (neuro-)degenerative, kidney, and gynecologic diseases.

## 2. Optoacoustic Imaging across Scales

A major advantage of OAI is its scalability; OAI can be performed over a wide range of depths and resolutions enabling molecular imaging across scales and increasing its applicability both experimentally and clinically [23,24]. While most imaging setups must adhere to the maximum permissible exposure (MPE) for the use of lasers in medicine as set by American National Standards Institute (ANSI) [25], OAI does not expose subjects to the risks associated with ionizing radiation.

While the first systems were developed as experimental imaging modalities for higher magnification and animal imaging, they have advanced to preclinical and clinical approaches that are mostly classified as microscopic, mesoscopic, or macroscopic/tomographic techniques (Table 1) [26,27,28]. Similar OAI technologies can deliver very different imaging information; the resolution scales linearly with penetration depth, so while optical resolution photoacoustic microscopy (OR-PAM) achieves less penetration depth but high lateral and spatial resolution of up to 1 µm by tight optical focusing, acoustic resolution photoacoustic microscopy (AR-PAM) has limited lateral and spatial resolution by loose acoustic focusing but a penetration depth up to several centimeters [29,30,31,32]. Optoacoustic mesoscopy (OPAM) is a modality which bridges the gap between microscopy and macroscopy, with acoustic resolution and wide bandwidth US detection for imaging volumes up to several millimeters in depth [33]. For preclinical imaging, ring detectors can be used to utilize loose acoustic focusing for the acquisition of whole-body data sets from living mice within 15–30 min [34,35,36]. In general, in acoustic resolution methods, the resolution of the system is determined by the acoustic detection components, whereas in optical resolution methods, the resolution is determined by the properties of the excitation light [37].

## 3. Acute and Chronic Inflammation

Inflammation is a broad topic impacting both physiological and pathophysiological processes in the human body [38]. Inflammation can not only promote neurodegeneration [39], metabolic disorders [40], allergic inflammation [41], or cancer [42], it is also required for the well-coordinated process of tissue repair and healing [43]. Whether the inflammation is initiated by thermal injury, trauma, infection, or other stimuli, OAI holds promise to visualize inflammation at different steps between the acute and chronic stages derived from cellular, vascular, or extracellular responses [44,45]. OAI has a unique ability to image many of the endogenous molecules present in the inflammatory process, such as hemoglobin and collagen (Figure 1), as well as label other inflammatory cells or molecules using exogenous contrast agents [6,9,27,36,46]. A brief overview of the inflammatory process and possible imaging targets is given in Figure 2.

## 4. Optoacoustic Imaging of Inflammation: Applications

The following section discusses translational and clinically relevant applications of OAI inflammation imaging. Table 2 gives an overview of selected studies.

### 4.1. Cardiovascular Diseases

A traditional domain for OAI has been the imaging of strong endogenous molecules such as hemoglobin, which is a dominant absorber in the near-infrared window I (NIR-I, 600–950 nm, Figure 1) [26]. Using multi-wavelength approaches, OAI systems are capable of distinguishing deoxygenated hemoglobin (Hb_R_, which is the dominant absorber at 700 nm) and oxygenated hemoglobin (HbO_2_, which is the dominant absorber at 850 nm). Total hemoglobin (Hb_T_ = Hb_R_ + HbO_2_) can be calculated as well as oxygenation levels (ratio of HbO_2_ to Hb_T_) [101]. In the visible and NIR-spectrum, there are also multiple isosbestic points (390, 422, 452, 500, 530, 545, 570, 584, and 797 nm), where signals reflect total hemoglobin (Hb_T_), regardless of the hemoglobin oxygenation level [31,102,103]. Owing to this capability, OAI has been widely used to visualize blood vessels and microcirculation, and to measure blood oxygenation levels, both experimentally in multiple organs [93,104] and clinically in patients [105]. Additionally, OAI has the potential to image blood flow velocity, although this has proved challenging at clinically relevant depths due to poor velocity signal-to-noise ratio [106,107,108]. Galanzha et al. demonstrated that a photoacoustic system is capable of monitoring multiple blood rheology parameters, such as red blood cell (RBC) aggregation, deformability, shape, intracellular hemoglobin distribution, individual cell velocity, hematocrit, and shear rate [106]. It could also enable diagnostics of infectious pathogens, such as bacteria, viruses, fungi, protozoa, parasites, and helminthes, or endogenous cells such as metastatic, infected, inflamed, stem, and dendritic cells [109,110].

When imaging the circulatory system in the clinic, a hand-held OAI device with a 128-element transducer array at a center frequency of 8 MHz was capable of depicting major blood vessels and microvasculature together with information about hemoglobin oxygen saturation and pulsation [59]. Similar systems demonstrated the ability to resolve vessels as small as 100 µm in diameter and within 1 cm depth, which could aid in accurately diagnosing vascular malformations [60]. Further studies were performed for mapping human carotid arteries [61] and monitoring altered hemoglobin signals and oxygenation in patients with peripheral artery disease [62]. These macroscopic approaches to visualize and monitor the human circulatory system using OAI could be translated to detect inflammatory hyperemia in the future.

OA cardiovascular imaging to date has mostly focused on detecting atherosclerosis and related diseases. While immune cells dominate early atherosclerotic lesions [111], developed atherosclerotic lesions, i.e., atheromata, are asymmetric focal thickenings of the innermost layer of an artery, consisting of inflammatory cells, connective-tissue, lipids, and debris [51,112].

In order to visualize these chronic inflammatory processes, imaging of chronic lipid deposition via OAI was proposed as an imaging biomarker to detect thromboembolism, and subsequently, to monitor the risk of stroke and myocardial infarction. Following this idea, Sangha et al. used a hybrid US/OAI system with illumination in the extended NIR range (1210 nm) to visualize the differences in lipid accumulation in murine periaortic fat according to gender, genotype, or age [51]. A more recent study described longitudinal dynamics of murine vascular remodeling as well as intraplaque hematoma and lipid deposition in a murine model of atherosclerosis using hybrid US/OAI at 1100 nm and 1210 nm [52].

In contrast to these transabdominal imaging approaches described above, several studies have also utilized catheter-based approaches ex vivo [47,53,54], in animal models [48,55] or human samples [49,50], with laser illumination around 930 nm for detecting the peak absorption of lipids in the first NIR window (Figure 1). Other studies aimed to increase the specificity of OAI for identification of carotid atherosclerosis by targeted approaches. An example is a study which used CD36-labeled semiconducting polymer nanoparticles in combination with OAI to identify the extent of the inflammation of carotid atherosclerosis in mice [56]. CD36 is a membrane glycoprotein present on various inflammatory cell types including monocytes, macrophages, endothelial cells, adipocytes, and platelets, which are all significantly involved in atherogenic processes [113]. Similar to this idea, Xie et al. used near-infrared erythrocyte-derived transducers (NETs), which are excited at 800 nm [57]. This study identified atherosclerosis by vascular occlusion due to accumulation and retention of NETs within the inflammatory lesions [57]. Wang et al. utilized gold nanoparticles, which cause a redshift in the optical absorbance spectra upon aggregation in macrophages of atherosclerotic plaque lesions [58]. In summary, OAI has the potential to deliver both anatomic (vascular) information alongside functional and targeted imaging signals providing biomarkers for increased cardiovascular risk. These studies further demonstrate the flexible use of OAI, which provides imaging insights into a multitude of biomarkers at various anatomical locations, depths, and resolutions. In comparison to traditional cross-sectional imaging modalities, OAI may provide the capabilities to directly visualize lipid-rich plaques, intraplaque hemorrhage or effects of therapeutic agents against plaques, or in-stent inflammation in a non-invasive manner [62]. Cardiovascular diseases are a major cause of mortality, meaning that earlier detection and improved monitoring of disease progression using OAI could have great clinical impact [114].

### 4.2. Dermatologic Diseases, Wound and Infection

In contrast to imaging deep vasculature, in dermal OAI, the trade-off between imaging resolution and depth shifts in favor towards imaging in higher resolution at superficial locations. Therefore, mesoscopic technologies such as raster-scanning optoacoustic mesoscopy (RSOM) are sufficient to visualize several millimeters into the skin and can be used to gain insight into dermatologic diseases [33]. An example of microvascular imaging derived from our own RSOM data is presented in Figure 3. In a preclinical context, dual-wavelength (532 nm and 559 nm) OAI was used to monitor the reperfusion of blood vessels and oxygen saturation changes in both arteries and veins in mouse ears to classify ischemic and unaffected tissue [115]. In a clinical proof-of-concept study, Berezhnoi et al. showed that RSOM could visualize the dilation of individual vessels in the skin microvasculature in response to local heating in a region of interest measuring 4 mm (x) × 2 mm (y) × 0.75 mm (z), using a 532 nm laser, an isosbestic point of hemoglobin [69]. This underlines the possible capability to map local hyperemia in skin as a biomarker for acute vascular response and inflammatory activity. Moreover, RSOM has been shown to visualize capillary loops and the horizontal vascular plexus in the skin, together with the ability to identify different dermal and epidermal layers [70]. This study suggested that a precise localization and visualization of characteristic inflammatory patterns is possible. Given this, the investigators demonstrated that RSOM could detect dilated and contorted blood vessels in the tips of the dermal papillae, as seen in psoriasis [71]. Subsequently, changes in vascularity and visualized hyperemia showed good correlation with clinical scores of disease activity in psoriasis patients [72]. Further studies in atopic dermatitis confirmed that it is possible to score inflammatory activity [73] and response to antibody therapy using RSOM imaging of skin layers and microvasculature [74]. Automation of feature detection and the inclusion of artificial intelligence may even improve objective diagnostics in RSOM and OA clinical imaging [116]. In the future, multi-wavelength RSOM may add additional functional information about other endogenous absorbers such as melanin content or blood oxygenation in different compartments of the human skin [117].

Chronic skin wounds, resulting from diabetes or occlusive peripheral arterial disease, pose a clinically interesting target [118,119]. Similar to thermal injuries, which have been previously imaged with optoacoustics [63,64,65], OAI could help to monitor healing and scar formation, and to guide further surgical interventions. In this regard, Wu et al. demonstrated with macroscopic OAI that burn wounds with a higher severity have greater tissue hypoxia and slower sO_2_ changes, while microscopic OAI revealed the onset of angiogenesis together with changes in blood vessel density [120]. In the clinic, wound (super-)infection and inflammation can complicate treatment and patient outcomes. Therefore, the effective identification and monitoring of vascular structures and micro-environment changes due to bacterial inflammation is paramount [121]. In this regard, a preclinical study found that lipopolysaccharide (LPS)-induced inflammatory responses of the microvasculature are highly dependent on the applied LPS concentrations [66]. This correlation suggests that not only the severity of ischemia, but also the risk for inflammation and infection could be quantified using OAI.

Additionally, the proximity of the target structures in skin and wound imaging enables topic application of contrast agents. An elegant study demonstrated the delivery of bacterial-specific imaging probes coupled to complex sugars, which are taken up in larger quantities by microbial organisms [67]. In a comparable approach, an OAI detectable fluorescent derivative of maltotriose (Cy7-1-maltotriose) was used to monitor the effectiveness of antibiotic treatment in *E. coli*-induced myositis and *S. aureus* wound infection models [68]. This could enable the rapid identification of bacterial colonization or infection burden earlier than classical cultivation techniques requiring up to several days for diagnosis. This may be a promising approach to also monitor complicated wound infection therapies before aiming for surgical interventions.

Taken together, these studies demonstrate the ability of multi-scale OAI, particularly novel RSOM techniques, to visualize and monitor skin alterations in multiple inflammatory dermatologic diseases. Going forward, OAI may be implemented as a tool to non-invasively monitor pharmacologic interventions and early response to therapy in these patients.

### 4.3. Gastrointenstinal Diseases

Similar to dermal inflammatory processes, mucosal inflammation and gastrointestinal diseases are well characterized by different OAI approaches (Figure 4). In animal models of experimental colitis, increased hemoglobin signals in inflamed intestinal walls were shown [122]. In humans, monitoring hemoglobin levels in intestinal walls with a hand-held OAI probe was clinically used to grade disease activity in patients with Crohn’s disease (CD) [85]. Increased signal levels of hemoglobin, displaying intestinal hyperemia, correlated well with endoscopic and histologic active disease and could therefore be used to differentiate between active disease and remission in a non-invasive fashion [84,85]. The device has since been developed further and a subsequent multicenter trial aims to determine whether the technique can be implemented in the clinical routine (https://euphoria2020.eu, last access: 9 April 2021, ClinicalTrials.gov Identifier: NCT04456400). A major application of this device would be the monitoring of anti-inflammatory regimes and simultaneously reducing the need for more invasive procedures, such as endoscopies (Figure 4A). In confocal laser endomicroscopy, fluorescently labeled TNFα [123] and α4β7 integrin [124] antibodies have been used to provide specific molecular markers of Crohn’s disease. Similar OA probes could be used in the future [66] to increase target specificity when monitoring intestinal inflammation.

Intestinal strictures pose a significant burden in chronic inflammatory bowel disease, where patients are often subjected to invasive endoscopic intervention or even surgery [125,126]. Yang et al. demonstrated a photoacoustic and ultrasonic dual-mode endoscope with the ability to image internal vasculature and lymphatic vessels in vivo [83]. OAI integrated into an acoustic resolution side-view endoscope with multi-wavelength imaging gave insight into hemoglobin and collagen content in intestinal strictures [127]. Even though current conventional cross-sectional imaging techniques are not sufficiently accurate for use in routine clinical practice, transabdominal imaging to quantify the burden of intestinal strictures is highly desirable [128] because it is less invasive. Imaging tissue scars and the proliferation of extracellular components could give insight into chronic inflammation in intestinal strictures. Conventional imaging modalities can identify the strictures but often cannot distinguish whether a stricture is predominantly inflammatory or fibrotic. An internal and external OAI approach was capable of visualizing both in a rat model of Crohn’s disease [80]. Even in surgically removed human intestinal stricture specimens, similar changes were found [81]. At mesoscopic levels, imaging of an inflamed colon by assessing the vascular architecture [129] and expansion of broad-absorbing tdTomato expressing activated tumor-associated fibroblasts in colorectal neoplasia was achieved [130].

Another interesting approach awaiting clinical translation is the OAI of acute and chronic liver damage together with liver fibrosis. Liver function assessed by indocyanine green (ICG) clearance and subsequently imaged with optoacoustics was used as a measure for acute acetaminophen-induced hepatotoxicity [75]. Additionally, herbal medicine-induced damage could be graded by a liposomal nanoprobe consisting of a NO responsive dye and a diketopyrrolopyrrole-based conjugated polymer [76]. Similar probes, either for detection of H_2_S or leucine aminopeptidase, were reported to be detectable by OAI after undergoing absorption redshifting [77,78]. The generated prominent optoacoustic signals provided measures for acute liver injury. Another study demonstrated that OAI can provide valuable information about ongoing chronic liver inflammation. In a mouse model of CCl_4_-induced liver fibrosis, OAI signals significantly increased in fibrotic livers when compared to healthy livers [79]. Interestingly, a single wavelength at 808 nm seemed to indicate collagen I in large concentrations around fibrotic nodules in murine livers. Even intraoperative imaging, for guiding surgical interventions along major vessels in the liver and pancreas, could be achieved with OAI [82]. A potential application to image changes in the composition of liver tissue is demonstrated in Figure 4B.

**Figure 4 biomedicines-09-00483-f004:**
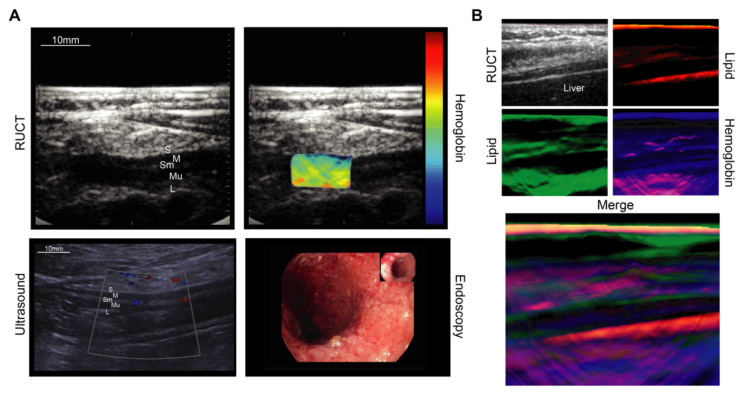
Multispectral optoacoustic tomography (MSOT) of intestinal inflammation using hemoglobin as a surrogate for disease activity. The technique enables hybrid imaging together with B-mode ultrasound imaging (RUCT). (**A**) MSOT imaging of the liver in humans enabling the generation of signals for different subcellular compounds; (**B**) S = serosa, M = muscularis mucosa, Sm = submucosa, Mu = mucosa, L = lumen, RUCT = reflectance ultrasound computed tomography. Figure (**A**) reproduced and modified from [131]. (https://creativecommons.org/licenses/by/4.0/ (accessed on 9 April 2021)).

### 4.4. Musculoskeletal Diseases

Rheumatoid arthritis (RA) is a chronic autoimmune disease leading to articular destruction associated with co-morbidities in vascular, metabolic, bone, and psychological domains due to tissue remodeling and damage [132,133]. In an RA animal model, an L-selectin/P-selectin-labeled contrast agent (a polyanionic dendritic polyglycerol sulfate (dPGS) labeled with an NIR fluorophore), which binds vascular leucocyte adherence proteins, enabled the accurate diagnosis of inflammation in the mouse joints using OAI [86]. Similarly, Fournelle et al. used an antitumor necrosis factor-α (TNF-α) antibody-coupled to gold nanorods as nanoprobes for confirming the overexpression of TNF-α in arthritic knees [87]. Instead of using nanoparticles, dye-VEGF-antibody-loaded microbubbles enabled multimodal OAI and US imaging of rats with inflammatory arthritis [88].

Additionally in psoriatic arthritis, the imaging of hemoglobin species with OAI may allow for the early detection of inflammation to prevent a delayed diagnosis and the associated complications [89]. Another study found that OAI signals acquired at 850 nm are sensitive to depicting increased blood volume indicating enthesis inflammation [90]. In the subcutaneous finger tissue of patients with systemic sclerosis (SSc), OAI signals for oxygenated hemoglobin and total hemoglobin reflected chronic microvascular dysfunction as a possible biomarker of disease activity [60]. Together with high-frequency US, OAI may distinguish between early and later forms of SSc by the means of tissue oxygenation and skin thickening [91]. At the mesoscopic level, multi-wavelength RSOM could visualize microvascular structure and oxygenation in murine paws (Figure 5), which may provide further insight into rheumatic diseases with future options for monitoring inflammation and the corresponding therapeutic interventions.

### 4.5. (Neuro-)Degenerative Diseases

Optoacoustic microscopy (OAM) and tomographic approaches have been recognized as tools to study animal models of brain diseases [134]. An example of utilizing contrast agents for imaging inflammation in the brain is the development of a CDnir7 probe for macrophage uptake [135]. Park et al. used this approach to visualize inflammation related to Alzheimer’s disease (AD) changes by visualizing more CDnir7 in the AD mouse cerebral cortex compared to that of normal mice [92]. Another study demonstrated the potential of utilizing dual-wavelength OAI (at 750 nm and 850 nm) to monitor cerebrovascular damage and hypo-oxygenation following brain injury, which is increasingly linked to neurodegenerative disease [93].

A first-in-human pediatric study showed that OAI can be used to assess the molecular composition of muscles in children. The inherited degenerative and fatal Duchenne muscular dystrophy is a disease usually treated by anti-inflammatory steroids. A hand-held US/OAI system was able to measure signals between 680 and 1100 nm correlating with fibrotic content as a novel non-invasive biomarker for disease severity and a potential monitoring tool for novel therapies (Figure 6) [94]. From a clinical perspective, this may open up the possibility of therapeutic monitoring at an earlier age and improve upon the current subjective functional testing or the use of other invasive imaging procedures.

The technology reliably measures OA signals with minimal error and excellent intra- and inter-operator agreement [136]. Similar to gastrointestinal applications, this first-in-human study frames the idea that it is possible to image late-stage chronic degenerative and inflammatory processes driven by fibrosis with OAI.

### 4.6. Kidney Diseases

In a mouse model, 5.5 nm-sized gold nanoparticle (GNP)-based probes were shown to be able to diagnose acute kidney injury by measuring positive OAI signals between 680 and 970 nm in the bladder of mice with temporal unilateral ureteral obstruction for 96 h [96]. This approach may open up a probe-based option for the early detection of inflammatory kidney injury.

Using ICG as a clinically approved contrast agent, preclinical OAI was capable of quantitatively measuring kidney perfusion [95] at 800 nm (Figure 7). Besides visualizing pharmacokinetic profiles, this approach may also be suitable for determining acute renal injuries and inflammation. In this regard, a more recent study demonstrated how OAI can be used to grade organ quality for transplantation by assessing kidney fibrosis as a marker of organ function in mice, pigs, and humans [97]. The authors even demonstrated that OAI can be performed in a human kidney transplantation-like setting, which suggests the emerging potential for clinical translation.

### 4.7. Gynaecologic Imaging

Clinical interventions in early life include monitoring placental function. In pre-eclampsia, a syndrome of impaired vasculature function, immune modulatory natural killer (NK) cells accumulate in the placenta [137]. OAI could visualize reduced placental perfusion in a rat model of pre-eclampsia [98]. Another interesting approach to monitor placental function was shown by Huda et al., who generated a folate-conjugated ICG contrast agent to image folate accumulation in the placenta [100].

The ability to directly visualize the function of this organ will foster the development of novel therapeutics to treat placental ischemia and halt the progression of pre-eclampsia [99].

## 5. Limitations of OAI

Despite its remarkable potential to image inflammation in a clinical setting, OAI has several limitations. Until now, penetration depths have still been limited to a few centimeters (up to ~7 cm), which limits the imaging of organs at depth, especially in patients with a high body mass index (BMI). However, not only increased body fat, but also the natural heterogenous fat distribution in different genders is a limiting factor [136]. OA endoscopy approaches may bypass this depth limitation in some scenarios. In addition, patients with increased hair or darker skin (skin type 4 to 6 on the Fitzpatrick scale) present a challenge due to the high level of light absorption by melanin, which will significantly reduce the light energy (fluence) reaching deeper tissue. Comparable problems are seen in animals with melanin expression (e.g., C57BL6 and others), limiting options for in vivo animal models with intact immune systems. The problem of wavelength- and depth-dependent decreases in light fluence distribution becomes more pronounced in deep tissues, causing errors in estimating blood oxygen saturation differences [46] in particular. Correcting for spectral distortions of illumination light as it passes through tissue would allow the absolute quantification of optoacoustic molecules, but it remains a significant challenge to apply in vivo and hence is an active area of research in the field [46,138].

A topic currently under investigation is the standardization of OAI measurements, especially in quantitative measurements for clinical applications [139,140]. Even though there is evidence for repeatable and stable OAI measurements over time [136], their use is still far from clinical “routine”. To aid clinical translation, the clinician would also require more intuitive approaches to interpret optoacoustic images, preferably together with sufficient familiar anatomical information. Therefore, a hybrid imaging approach together with co-registration of (high-resolution) B-mode US signals is highly important [141,142]. The use of exogenous contrast agents to specifically label target molecules should be conducted with caution, owing to the possible toxicities associated with their administration [143].

Finally, most of the clinical studies conducted to date have been observational and/or conducted on a limited number of patients. The studies have not impacted the clinical decision-making of the patient. Therefore, the true impact and benefits of using OAI in clinical inflammation imaging need to be assessed in multi-center prospective studies, of which there is one study currently underway in Europe (https://euphoria2020.eu, last access: 9 April 2021, ClinicalTrials.gov Identifier: NCT04456400).

## 6. Conclusions

OAI is an emerging non-invasive imaging modality which has already had significant impact on unravelling the pathophysiology of inflammation in preclinical and clinical scenarios. OAI provides a multi-scale perspective of inflammatory processes, monitoring both structural and functional aspects at various anatomical locations, resolutions, and imaging depths. Besides visualizing strong absorbing hemoglobin, and therefore vascular structure, future approaches will focus on novel endogenous, as well as exogenous, OA compounds.

As shown by the first meaningful clinical studies described here, OAI may be sensitive enough to detect even slight inflammatory changes in tissues, therefore enabling early or even preventive treatment strategies. Despite the first preclinical studies, OAI awaits the translation of novel contrast agents for imaging inflammatory cells or microorganisms [144]. This approach could also enable the targeted delivery of functionalized multimodal and theranostic compounds to abnormal tissues while sparing the other organ systems [145,146,147]. At the same time, a high signal intensity and a high degree of biodegradation is required when administered in humans [148].

## Figures and Tables

**Figure 1 biomedicines-09-00483-f001:**
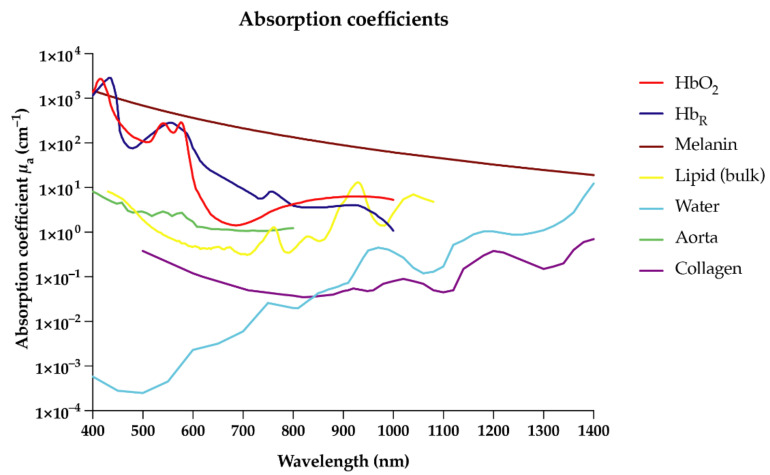
Absorption coefficients (µ_a_ in cm^−1^) versus wavelength (in nm) for different optoacoustic imaging molecules and tissue. Spectra were derived from existing data as indicated: melanin (https://omlc.org/spectra/melanin/mua.html as derived from [15,16,17,18]), oxy- (HbO_2_) and deoxyhemoglobin (Hb_R_) (https://omlc.org/spectra/hemoglobin/summary.html), (bulk) lipid (https://omlc.org/spectra/fat/fat.txt as derived from [19]), water (https://omlc.org/spectra/water/data/hale73.txt derived from [20]), aorta tissue (https://omlc.org/spectra/aorta/oraevsky_a.txt derived from [21]), and collagen (extracted from [22]). All databases accessed on 9 April 2021.

**Figure 2 biomedicines-09-00483-f002:**
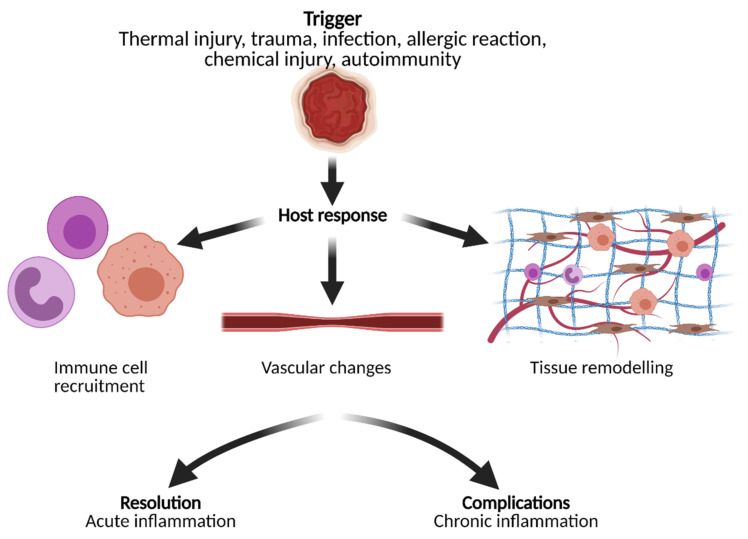
Inflammation is a multistep process triggered by a variety of causes and agents. This results in acute and/or chronic inflammation, which leads to host response in terms of cell recruitment, adaptions of the vasculature, and changes of the tissue composition and extracellular matrix. All of these aspects pose possible targets for opto-/photoacoustic imaging. Figure created with BioRender.com.

**Figure 3 biomedicines-09-00483-f003:**
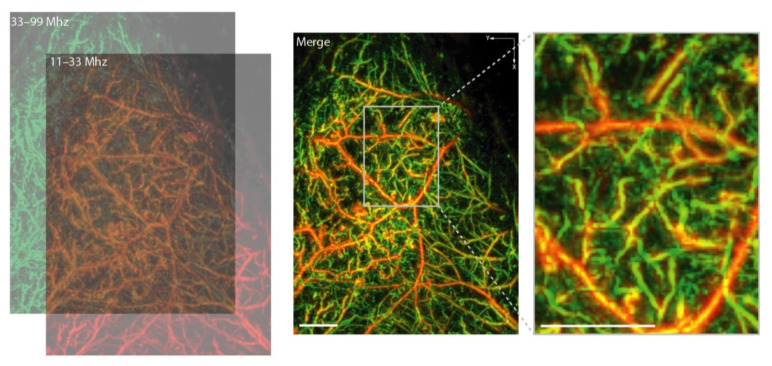
Raster-scanning optoacoustic mesoscopy enables a precise visualization of microvasculature. Depending on the detection frequencies, larger (11–33 MHz, in red) or smaller (33–99 MHz, in green) vessels can be resolved. White square indicates magnified area. Scale bar indicates 1 mm.

**Figure 5 biomedicines-09-00483-f005:**
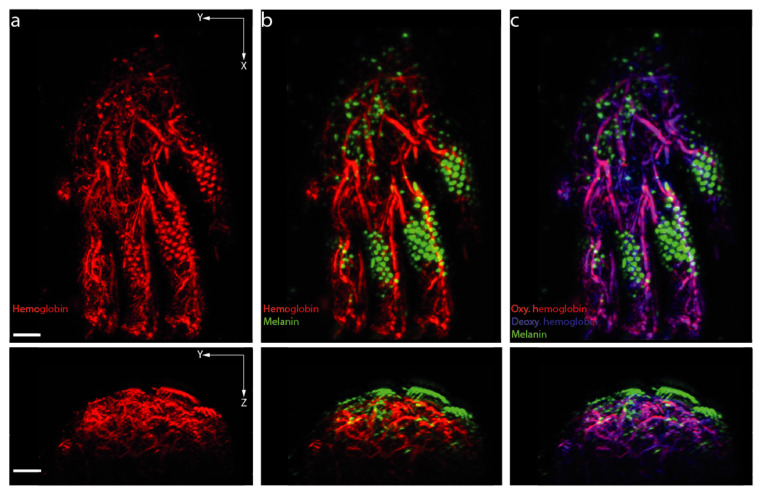
Raster-scanning optoacoustic mesoscopy (RSOM) of murine paw vasculature and the potential imaging readouts for inflammation. (**a**) Shows the hemoglobin signal representing the vascular network, which can be used to identify changes in blood volume; (**b**) multi-wavelength illumination enables the separation of melanin and hemoglobin signals; (**c**) further unmixing enables the visualization of oxygenation status of hemoglobin. Scale bar indicates 1mm.

**Figure 6 biomedicines-09-00483-f006:**
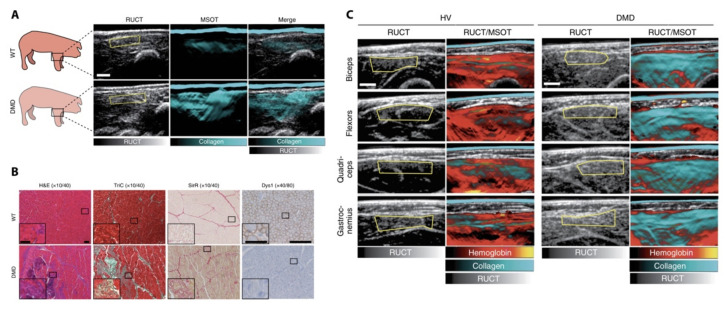
Optoacoustic imaging (OAI) of muscle degeneration in Duchenne muscular dystrophy. (**A**) Shows signals derived from wild-type (WT) and Duchenne muscular dystrophy (DMD) transgenic pigs; (**B**) demonstrates ex vivo tissue changes and expansion of extracellular matrix and collagens in diseased tissues as a possible correlate of OAI signals; (**C**) signals derived from healthy volunteers (HV) and pediatric patients with DMD at different anatomical positions. RUCT and RUCT/MSOT merged images shown. RUCT = reflectance ultrasound computed tomography, MSOT = multispectral optoacoustic tomography. Signals unmixed for hemoglobin and collagen. Figure modified with permission from [94]. This image is not published under the terms of the CC-BY license. For permission to reuse, please see [72].

**Figure 7 biomedicines-09-00483-f007:**
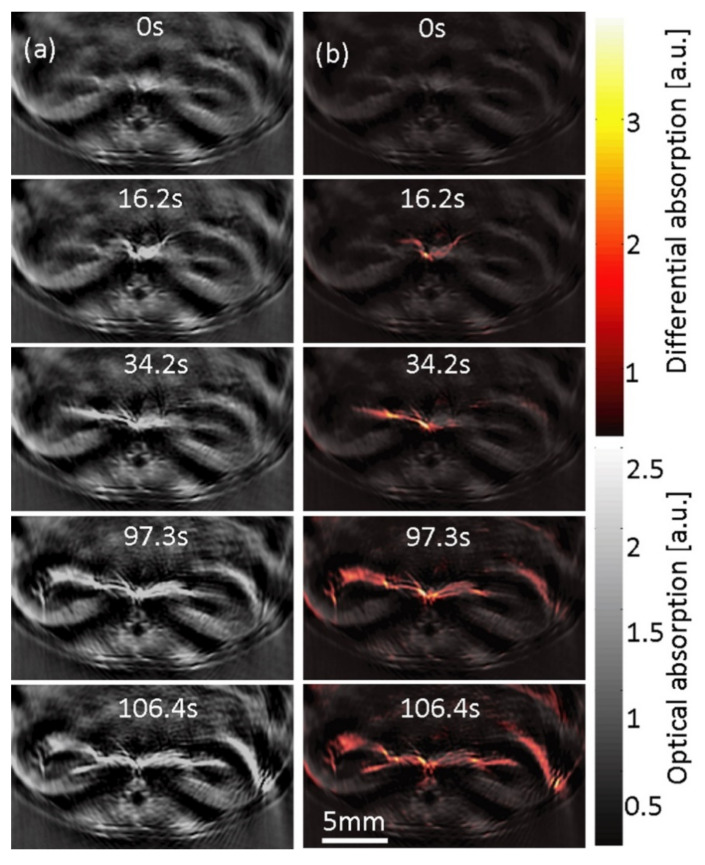
Tomographic optoacoustic imaging of indocyanine perfusion kinetics in murine kidneys. (**a**) Cross-sectional optoacoustic images over time of mouse kidneys at 800 nm after ICG injection; (**b**) the absorption difference with the single-wavelength image before injection to show increased ICG perfusion over time. Adapted with permission from [95]. This image is not published under the terms of the CC-BY license. © The Optical Society.

**Table 1 biomedicines-09-00483-t001:** Optoacoustic imaging (OAI) technologies across scales (US = ultrasound, NIR = near-infrared range, exNIR = extended near-infrared range).

Technique		Resolution	Penetration Depth	Laser Source	Focus	References
Microscopy	Optical resolution	~1 µm ~85 µm	>1 mm	Visible/NIR	US < Optical	[30]
	Acoustic resolution		~3 mm	Visible/NIR	US > Optical	
Mesoscopy		~10 µm	1–4 mm	Visible	US > Optical	[33]
Macroscopy/ Tomography		~250 µm	2–5 cm	NIR/exNIR	None	[36]

**Table 2 biomedicines-09-00483-t002:** Selected inflammatory diseases studied with OAI. ICG = indocyanine green, NET = near-infrared erythrocyte-derived transducers, NP = nanoparticle, VEGF = Vascular endothelial growth factor, TNF α = tumor necrosis factor-α.

Diseases	Condition	Stage	Target/Contrast	Reference
Cardiovascular	Atherosclerosis	Preclinical	Lipid	[47,48,49,50,51,52,53,54,55]
	Atherosclerosis	Preclinical	CD36 targeted NP	[56]
	Atherosclerosis	Preclinical	ICG loaded NETs	[57]
	Atherosclerosis	Preclinical	Gold nanoparticles	[58]
	Foot vasculature	Clinical	Hemoglobin	[59]
	Vascular malformations	Clinical	Hemoglobin	[60]
	Carotid arteries	Clinical	Hemoglobin	[61]
	Peripheral artery disease	Clinical	Hemoglobin	[62]
Dermatologic	Thermal injuries	Preclinical	Hemoglobin	[63,64,65]
	LPS-induced wound inflammation	Preclinical	Hemoglobin	[66]
	Bacterial wound infection	Preclinical	Targeted sugars	[67,68]
	Skin microvasculature, layers	Clinical	Hemoglobin	[69,70]
	Psoriasis	Clinical	Hemoglobin	[71,72]
	Atopic dermatitis	Clinical	Hemoglobin	[73,74]
Gastrointestinal	Acute liver damage	Preclinical	ICG Perfusion	[75]
	Acute liver damage	Preclinical	Probes, NO, H2S and leucine aminopeptidase	[76,77,78]
	Liver fibrosis	Preclinical	Collagen	[79]
	Intestinal strictures	Preclinical	Collagen	[80,81]
	Intestinal inflammation	Preclinical	Hemoglobin	[82,83]
	Image-guided surgery	Preclinical	Hemoglobin	[82]
	Intestinal vasculature and lymphatic vessels	Preclinical	Hemoglobin, Evans blue dye	[83]
	Intestinal inflammation	Clinical	Hemoglobin	[84,85]
Musculoskeletal	Arthritis	Preclinical	NP: targeting L-selectin/P-selectin, TNFα, VEGF	[86,87,88]
	Rheumatoid arthritis	Clinical	Hemoglobin	[89]
	Enthesitis	Clinical	Hemoglobin	[90]
	Systemic sclerosis	Clinical	Hemoglobin	[60,91]
Neurodegenerative	Alzheimer’s diseases	Preclinical	CDnir7	[92]
	Cerebrovascular damage	Preclinical	Hemoglobin	[93]
	Muscular dystrophy	Clinical	Collagen	[94]
Kidney	Organ perfusion	Preclinical	ICG Perfusion	[95]
	Acute injury	Preclinical	NP	[96]
	Organ transplant	Clinical	Collagen	[97]
Gynecologic	Preeclampsia	Preclinical	Hemoglobin	[98,99]
	Preeclampsia	Preclinical	ICG targeting FRα	[100]

## Data Availability

Not applicable.

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
