# Peer review of "Optoacoustic Imaging in Inflammation"

_biomedicines, 2021, doi:10.3390/biomedicines9050483_

Round 1

Reviewer 1 Report

Summary: This review focuses on the application of optoacoustic imaging (OAI) in preclinical and clinical settings for evaluation of inflammatory conditions ranging from cardiovascular disease to musculoskeletal disorders. The authors make a distinction between utilizing endogenous and exogenous chromophores as biological markers to characterize inflammation as well as the scalability of OAI (microscopic, mesoscopic, and macroscopic). While this review does a nice job comparing different OAI approaches for a variety of applications, there are a few major concerns that should be addressed before publication. For example, a more extensive discussion on the limitations and/or need for improvement of OAI technology for each application would be helpful, as well as information surrounding the development of therapeutics to better fit the aims of the journal.

Major Comments:

  1. An expansion upon the limitations of OAI for each application should be included either within each application section or as part of the conclusion. What is preventing OAI from being widely used in the clinic? Highlighting practical limitations can allow readers to identify and solve problems that could lead to advancements in the field. For example, stating that a strength of OAI is that “can be performed over a wide range of depths” could be true compared to microscopy techniques, but is almost certainly not true when compared to clinical ultrasound, CT, or MRI. Further, using OAI to image inflammation specifically requires contrast agents or other exogenous compounds.
  2. This manuscript mentions the possibility for therapeutics to be developed as a result of using OAI as a tool to assess inflammation. While it is an important topic mentioned throughout the manuscript, specifics regarding these therapies are not included. This would be helpful in the Conclusions section of the paper on page 10. What sort of novel therapies can be developed as a result of being able to monitor inflammation? How do they differ for each application? Providing this information is especially relevant to Biomedicines because the aims and scope of the journal emphasize “the discovery and characterization of new therapeutic targets, therapeutic strategies, and research of naturally driven biomedicines, pharmaceuticals, and biopharmaceutical products”.
  3. While impossible to reference every study on OAI imaging and noting the authors have already included 98 references, the majority appear to come from only a few well-known laboratories. Suggest additional references focused on OAI imaging of inflammatory-related diseases including PMIDs 23278229, 28270980, 30679723, 31411010, 32266325, 32442681, and 33585582.

Minor Comments:

  1. While OAI does not expose patients to the risks associated with other imaging modalities such as radiation exposure during an x-ray, there are still guidelines that must be followed to maintain patient safety. This includes the standards established by the American National Standards Institute (ANSI 1). This would be helpful to include and reference in the introduction.
  2. Typo in the introduction. Second paragraph, first sentence, “broad approach to image and understand” should be changed to “broad approach to imaging and understanding”.
  3. Typo on page 3. “…repair und healing (26).”
  4. Figure 1 caption incomplete. Did the authors mean to say, “reproduced with permission from (10)”?
  5. Table 1 “US” acronym needs to be explained in manuscript for those who do not know that this stands for “ultrasound”.
  6. Typo in section 4.1, paragraph 3, sentence 2, “are asymmetric focal thickenings of the innermost layer an artery” should be “are asymmetric focal thickenings of the innermost layer of an artery”.
  7. Referring to 930nm as the wavelength of “the peak absorption of lipids” is misleading on page 5. As Figure 1 shows, there is a higher absorption peak at 1210nm for lipids. Suggest adding clarification that 930nm is the peak within the first near infrared window.
  8. Typo in section 4.2, paragraph 1, sentence 2: add space between words in “aresufficient”.
  9. Random reference to an article (19) with no sentence preceding it in section 4.2, paragraph 1, before second to last sentence.
  10. Typo in section 4.2, paragraph 2: “In clinic, wound (super-)infection and inflammation…” needs to change to “In the clinic, wound (super-)infection and inflammation…”.
  11. Reference(s) missing for Figure 3 and Figure 5 in the captions. If it is the authors’ own work, then it should be mentioned in the figure caption or preceding the reference to this figure within the manuscript’s text.
  12. Page 6 – superfluous comma can be removed. “…demonstrated with macroscopic OAI, that burn wounds…”
  13. Typo in section 4.3, paragraph 3, last sentence: “…in composition of liver tissue in demonstrated in Figure 4B” needs to change to “…in composition of liver tissue is demonstrated in Figure 4B”.
  14. In Figure 4, what wavelengths were used to excite the tissues in these images and why are there two lipid panels in 4B?
  15. Typo in Figure 4 caption. “MSOT imaging of the liver in humans enabling to generate signals for different…” should be modified to something like “MSOT imaging of the liver in humans enabling generation of signals for different…”.
  16. Again, where is the reference for Figure 5?
  17. Suggest expanding sections 4.5 and 4.6 to include what wavelengths are used in measuring neurodegenerative and kidney diseases and why. These sections would benefit from the inclusion of images like the ones shown in Figure 5 to help elucidate the capabilities of OAI in these areas.

Author Response

Point to point reply

Reviewer 1:

Comment 1:

An expansion upon the limitations of OAI for each application should be included either within each application section or as part of the conclusion. What is preventing OAI from being widely used in the clinic? Highlighting practical limitations can allow readers to identify and solve problems that could lead to advancements in the field. For example, stating that a strength of OAI is that “can be performed over a wide range of depths” could be true compared to microscopy techniques, but is almost certainly not true when compared to clinical ultrasound, CT, or MRI. Further, using OAI to image inflammation specifically requires contrast agents or other exogenous compounds.

Response 1:

We thank the reviewer for this useful comment. We have included a paragraph on OAI limitations towards the end of the review (see paragraph 5), where we discuss penetration depth and light fluence distribution problems, OAI standardization, the use of contrast agents and highlight important concepts for clinical translation.

Comment 2:

This manuscript mentions the possibility for therapeutics to be developed as a result of using OAI as a tool to assess inflammation. While it is an important topic mentioned throughout the manuscript, specifics regarding these therapies are not included. This would be helpful in the Conclusions section of the paper on page 10. What sort of novel therapies can be developed as a result of being able to monitor inflammation? How do they differ for each application? Providing this information is especially relevant to Biomedicines because the aims and scope of the journal emphasize “the discovery and characterization of new therapeutic targets, therapeutic strategies, and research of naturally driven biomedicines, pharmaceuticals, and biopharmaceutical products”.

Response 2:

We agree with the reviewer and included an expanded text in paragraph 6.

Comment 3:

While impossible to reference every study on OAI imaging and noting the authors have already included 98 references, the majority appear to come from only a few well-known laboratories. Suggest additional references focused on OAI imaging of inflammatory-related diseases including PMIDs 23278229, 28270980, 30679723, 31411010, 32266325, 32442681, and 33585582.

Response 3:

We have referenced the proposed studies and reviews. Accordingly, we included numerous other (older) studies.

Minor

Comment 1:

While OAI does not expose patients to the risks associated with other imaging modalities such as radiation exposure during an x-ray, there are still guidelines that must be followed to maintain patient safety. This includes the standards established by the American National Standards Institute (ANSI 1). This would be helpful to include and reference in the introduction.

Response 1:

The follow sentence was included in section 2.:

“While most imaging setups must adhere to the maximum permissible exposure (MPE) for the use of lasers in medicine as set by American National Standards Institute (ANSI) (13), OAI does not expose subjects to the risks associated with ionizing radiation.”

Comment 2:

Typo in the introduction. Second paragraph, first sentence, “broad approach to image and understand” should be changed to “broad approach to imaging and understanding”.

Response 2:

Sentence was changed to: This review discusses the unique ability and scalability of opto- or photoacoustic imaging (OAI/PAI) to visualize, monitor and understand the molecular mechanisms of inflammation.

Comment 3:

Typo on page 3. “…repair und healing (26).”

Response 3:

Corrected.

Comment 4:

Figure 1 caption incomplete. Did the authors mean to say, “reproduced with permission from (10)”?

Response 4:

Corrected to “Absorption spectra for different endogenous optoacoustic imaging molecules reproduced with permission from (11).”

Comment 5

Table 1 “US” acronym needs to be explained in manuscript for those who do not know that this stands for “ultrasound”.

Response 5:

Included in table caption.

Comment 6:

Typo in section 4.1, paragraph 3, sentence 2, “are asymmetric focal thickenings of the innermost layer an artery” should be “are asymmetric focal thickenings of the innermost layer of an artery”.

Response 6:

Corrected.

Comment 7:

Referring to 930nm as the wavelength of “the peak absorption of lipids” is misleading on page 5. As Figure 1 shows, there is a higher absorption peak at 1210nm for lipids. Suggest adding clarification that 930nm is the peak within the first near infrared window.

Response 7:

Clarification included as proposed.

Comment 8:

Typo in section 4.2, paragraph 1, sentence 2: add space between words in “aresufficient”.

Response 8:

Corrected.

Comment 9:

Random reference to an article (19) with no sentence preceding it in section 4.2, paragraph 1, before second to last sentence.

Response 9:

Sentence included.

Comment 10:

Typo in section 4.2, paragraph 2: “In clinic, wound (super-)infection and inflammation…” needs to change to “In the clinic, wound (super-)infection and inflammation…”

Response 10:

Corrected.

Comment 11:

Reference(s) missing for Figure 3 and Figure 5 in the captions. If it is the authors’ own work, then it should be mentioned in the figure caption or preceding the reference to this figure within the manuscript’s text.

Response 11:

Comments included in captions/text.

Comment 12:

Page 6 – superfluous comma can be removed. “…demonstrated with macroscopic OAI, that burn wounds…”

Response 12:

Removed.

Comment 13:

Typo in section 4.3, paragraph 3, last sentence: “…in composition of liver tissue in demonstrated in Figure 4B” needs to change to “…in composition of liver tissue isdemonstrated in Figure 4B”.

Response 13:

Corrected.

Comment 14:

In Figure 4, what wavelengths were used to excite the tissues in these images and why are there two lipid panels in 4B?

Response 14:

Panel represented collagen signals. Labelling was corrected.

Comment 15:

Typo in Figure 4 caption. “MSOT imaging of the liver in humans enabling to generate signals for different…” should be modified to something like “MSOT imaging of the liver in humans enabling generation of signals for different…”.

Response 15:

Corrected.

Comment 16:

Again, where is the reference for Figure 5?

Response 16:

Comments included in captions/text. See comment 11.

Comment 17:

Suggest expanding sections 4.5 and 4.6 to include what wavelengths are used in measuring neurodegenerative and kidney diseases and why. These sections would benefit from the inclusion of images like the ones shown in Figure 5 to help elucidate the capabilities of OAI in these areas.

Response 17:

Wavelength information and images added.

Reviewer 2 Report

This Review “Optoacoustic imaging in inflammation” presented by Ferdinand Knieling’s team may have a high impact for investigators of inflammations.

The review is focused to analysis of opto- or photoacoustic imaging (OAI/PAI), a non-invasive visualization of laser-illuminated tissue by detection of acoustic signals in different inflammations. This review has the technical considerations of using OAI in current emerging translational applications.

Below there are some remarks:

This review doesn’t have a systematic analysis, novelty and even determined tasks; it has only the listing of published facts for the examples of inflammations.

The authors should make verification of the captions for the figures, and ask permissions from the publishers. For example, Figure 1.  Absorption spectra for different endogenous optoacoustic imaging molecules reproduced with from (10, it is Weber et al. 2016, but original figure was published in   Beard P. Interface Focus. 2011; 1(4): 602–31. (Ref # 13 is in list of reference).

Part “Optoacoustic imaging across scales”.

This part could be improved though analysis of OA and PA images of different modalities and their combinations with other imaging techniques to improve inflammation related imaging.

Reordering of this part would be redone for more strong demonstration of capabilities of different modalities of OAI. It is possible to reorganize (and add) part by part as follow:

Opto acoustic microscopy;

Opto acoustic sensing (1D and 2D);

Opto-acoustic imaging (2D);

Opto-acoustic tomography (3D);

Multispectral optoacoustic imaging and tomography;

Acoustic- optoacoustic imaging technology;

Combinations OAI with fluorescence and photothermal imaging (for example, the review Li K, Liu B. Chem Soc Rev. 2014 Sep 21;43(18):6570-97. doi: 10.1039/c4cs00014e).

This order can be used as a base for the analytical reviewing  of OAI capabilities for the acute and chronic inflammation and examples of applications.

For the part “Dermatologic diseases, wound and infection”

The captures for the Figure 3. Raster-scanning optoacoustic mesoscopy enables a precise visualization of microvasculature. It does not have no references, nor permissions.

Next paragraph has the similar problem.  Figure 4. Multispectral optoacoustic tomography (MSOT) of intestinal inflammation using hemoglobin as a surrogate for disease activity...  The figure is adapted from (93), it looks like cross reference, the figure should come from an original issue with the permission from publisher but not as copy/paste form other review (93).

This paragraph should include several lines about potential problems for the skin imaging: difficulties of light penetration and what kind of technique can  improve this imaging.

Figure 5. Raster-scanning optoacoustic mesoscopy (RSOM) of murine paw vasculature and potential imaging readouts for inflammation… in the paragraph 4.4. Musculoskeletal diseases does not have any reference.

Acknowledgments is not very clear: The authors acknowledged Ida Allabauer for excellent assistance during imaging. Is it experimental imaging, or assistance for adjustment images from published issues?

The authors have well presented data from teams of Dr’s Razansky D, and Ntziachristos V. In the same time, the international recognized labs and investigators are not listed well. There are:

Dr’s Emelianov S.Y; Gambhir S.S.; Wang L.V.; Zharov V.P (Galanzha EI, Zharov VP. Cytometry A. 2011 Oct;79(10):746-57. Review); Monohar S.; Liz-Marzan L. or Alexander Oraevsky group. Here are some examples of the references from his lab:

Longo, D.L.; Stefania, R.; Aime, S.; Oraevsky, A. Int. J. Mol. Sci. 2017, 18, 1719. Liopo A.V. and Oraevsky A.A. Nanoparticles as contrast agents for optoacoustic imaging. NJ, USA, John Wiley and Sons, 2015, Chapter 5: 111-149, ISBN: 978-1-118-12118-4; Conjusteau A., et al. Optoacoustic sensor for nanoparticle linked immunosorbent assay (NanoLISA), Proc. SPIE 7899, Progress Biomed Optics  Imaging, 2011, 12,17: 789910-19, doi:10.1117/12.879401

Review should be reorganized and the tasks should be strong formulated.

For sure, this review can be accepted for publication after the rewriting.

My decision is “Reconsider after major revision”

Author Response

Point to point reply

Reviewer 2

This review doesn’t have a systematic analysis, novelty and even determined tasks; it has only the listing of published facts for the examples of inflammations.

Comment 1:

The authors should make verification of the captions for the figures, and ask permissions from the publishers. For example, Figure 1.  Absorption spectra for different endogenous optoacoustic imaging molecules reproduced with from (10, it is Weber et al. 2016, but original figure was published in   Beard P. Interface Focus. 2011; 1(4): 602–31. (Ref # 13 is in list of reference).

Response 1:

We thank the reviewer for pointing this out. We have received permissions where necessary. For Figure 1, we have re-drawn the figure using spectra from this source: https://omlc.org/spectra/index.html

Comment 2:

Part “Optoacoustic imaging across scales”.

This part could be improved though analysis of OA and PA images of different modalities and their combinations with other imaging techniques to improve inflammation related imaging.

Reordering of this part would be redone for more strong demonstration of capabilities of different modalities of OAI. It is possible to reorganize (and add) part by part as follow:

Opto acoustic microscopy;

Opto acoustic sensing (1D and 2D);

Opto-acoustic imaging (2D);

Opto-acoustic tomography (3D);

Multispectral optoacoustic imaging and tomography;

Acoustic- optoacoustic imaging technology;

Combinations OAI with fluorescence and photothermal imaging (for example, the review Li K, Liu B. Chem Soc Rev. 2014 Sep 21;43(18):6570-97. doi: 10.1039/c4cs00014e).

This order can be used as a base for the analytical reviewing of OAI capabilities for the acute and chronic inflammation and examples of applications.

Response 2:

We thank the reviewer for the critical comment. The review was written with a specific focus on inflammation and discovering the biology behind it. A “technical” reorganization of the content would shift the focus. There are already numerous technical reviews (for example: https://www.journals.elsevier.com/photoacoustics/review-articles). Therefore, we did not rearrange the entire manuscript but expanded existing paragraph and included novel concepts (Limitations, gynecologic applications).

Comment 3:

For the part “Dermatologic diseases, wound and infection” 

The captures for the Figure 3. Raster-scanning optoacoustic mesoscopy enables a precise visualization of microvasculature. It does not have no references, nor permissions.

Next paragraph has the similar problem.  Figure 4. Multispectral optoacoustic tomography (MSOT) of intestinal inflammation using hemoglobin as a surrogate for disease activity...  The figure is adapted from (93), it looks like cross reference, the figure should come from an original issue with the permission from publisher but not as copy/paste form other review (93). 

This paragraph should include several lines about potential problems for the skin imaging: difficulties of light penetration and what kind of technique can improve this imaging.

Figure 5. Raster-scanning optoacoustic mesoscopy (RSOM) of murine paw vasculature and potential imaging readouts for inflammation… in the paragraph 4.4. Musculoskeletal diseases does not have any reference.

Acknowledgments is not very clear: The authors acknowledged Ida Allabauer for excellent assistance during imaging. Is it experimental imaging, or assistance for adjustment images from published issues?

Response 3:

 For Figures 3 and 5, we have added that the work is unpublished data from the authors. For Figure 4, this has been published under the creative commons license, which allows us to share and adapt the figure providing sufficient attribution to the original work is given. We have added a new paragraph called ‘Limitations of OAI’ to the end of the review to discuss difficulties in skin imaging and limitations with light penetration.  Acknowledgments have been updated.

Comment 4:

The authors have well presented data from teams of Dr’s Razansky D, and Ntziachristos V. In the same time, the international recognized labs and investigators are not listed well. There are:

Dr’s Emelianov S.Y; Gambhir S.S.; Wang L.V.; Zharov V.P (Galanzha EI, Zharov VP. Cytometry A. 2011 Oct;79(10):746-57. Review);

Monohar S.; Liz-Marzan L. or Alexander Oraevsky group. Here are some examples of the references from his lab:

Longo, D.L.; Stefania, R.; Aime, S.; Oraevsky, A. Int. J. Mol. Sci. 201718, 1719. Liopo A.V. and Oraevsky A.A. Nanoparticles as contrast agents for optoacoustic imaging. NJ, USA, John Wiley and Sons, 2015, Chapter 5: 111-149, ISBN: 978-1-118-12118-4; Conjusteau A., et al. Optoacoustic sensor for nanoparticle linked immunosorbent assay (NanoLISA), Proc. SPIE 7899, Progress Biomed Optics  Imaging, 2011, 12,17: 789910-19, doi:10.1117/12.879401

 Response 4:
We have included the Galanzha et al. reference kindly suggested by the reviewer in section 4.1. We have also expanded on literature directly relevant to inflammation imaging by citing the following studies: PMIDs 23278229, 28270980, 30679723, 31411010, 32266325, 32442681 and 33585582. Additionally, we included numerous other studies, however, some of the proposed citations are not within the scope of the current review.

Comment 5:

Review should be reorganized and the tasks should be strong formulated.

Response 5:

 See response 2 regarding reorganization and article focus.

Round 2

Reviewer 2 Report

This Review “Optoacoustic imaging in inflammation” presented by Ferdinand Knieling’s team may have a high impact for investigators of inflammations.

The review is focused to analysis of opto- or photoacoustic imaging (OAI/PAI), a non-invasive visualization of laser-illuminated tissue by detection of acoustic signals in different inflammations.

This review significant reorganized and looks like novel systematic analysis of a current published data in area of OAI/PAI. Manuscript has a novelty and good summarization of examples OAI of inflammations.

Acoustic- optoacoustic imaging technology;

However, I am continuing see that biggest part of reviving publications from teams of Dr’s Razansky D, and Ntziachristos V. and very interesting investigators

are not listed well.

There are:

Dr’s Emelianov S.Y; Gambhir S.S.; Wang L.V.; Zharov V.P, (Galanzha EI, Zharov VP. Cytometry A. 2011 Oct;79(10):746-57. Review); Monohar S.; Liz-Marzan L. or Alexander Oraevsky group. Here are some examples of the references from his lab:

Longo, D.L.; Stefania, R.; Aime, S.; Oraevsky, A. Int. J. Mol. Sci. 2017, 18, 1719.

Liopo A.V. and Oraevsky A.A. Nanoparticles as contrast agents for optoacoustic imaging. NJ, USA, John Wiley and Sons, 2015, Chapter 5: 111-149, ISBN: 978-1-118-12118-4;

Conjusteau A., et al. Optoacoustic sensor for nanoparticle linked immunosorbent assay (NanoLISA), Proc. SPIE 7899, Progress Biomed Optics  Imaging, 2011, 12,17: 789910-19, doi:10.1117/12.879401

I believe that some additional references are necessary for the manuscript.

For sure, this review can be accepted for publication with minor revision

Author Response

Reviewer:

This Review “Optoacoustic imaging in inflammation” presented by Ferdinand Knieling’s team may have a high impact for investigators of inflammations.

The review is focused to analysis of opto- or photoacoustic imaging (OAI/PAI), a non-invasive visualization of laser-illuminated tissue by detection of acoustic signals in different inflammations.

This review significant reorganized and looks like novel systematic analysis of a current published data in area of OAI/PAI. Manuscript has a novelty and good summarization of examples OAI of inflammations.

Acoustic- optoacoustic imaging technology;

However, I am continuing see that biggest part of reviving publications from teams of Dr’s Razansky D, and Ntziachristos V. and very interesting investigators are not listed well.

There are:

Dr’s Emelianov S.Y; Gambhir S.S.; Wang L.V.; Zharov V.P, (Galanzha EI, Zharov VP. Cytometry A. 2011 Oct;79(10):746-57. Review); Monohar S.; Liz-Marzan L. or Alexander Oraevsky group. Here are some examples of the references from his lab:

Longo, D.L.; Stefania, R.; Aime, S.; Oraevsky, A. Int. J. Mol. Sci. 201718, 1719.

Liopo A.V. and Oraevsky A.A. Nanoparticles as contrast agents for optoacoustic imaging. NJ, USA, John Wiley and Sons, 2015, Chapter 5: 111-149, ISBN: 978-1-118-12118-4;

Conjusteau A., et al. Optoacoustic sensor for nanoparticle linked immunosorbent assay (NanoLISA), Proc. SPIE 7899, Progress Biomed Optics Imaging, 2011, 12,17: 789910-19, doi:10.1117/12.879401

I believe that some additional references are necessary for the manuscript.

For sure, this review can be accepted for publication with minor revision

Response:

We thank the reviewer for his valuable comment and included a significant number of new citations from the following groups:

Wang: (1-5)

Oraevsky: (6-10)

Emelianov: (11-15)

Gambhir: (16-19)

Zharov: (20-22)

  1. Hu S, Wang LV. Photoacoustic imaging and characterization of the microvasculature. Journal of biomedical optics. 2010;15(1):011101.
  2. Wang LV, Hu S. Photoacoustic tomography: in vivo imaging from organelles to organs. Science. 2012;335(6075):1458-62.
  3. Yang JM, Favazza C, Chen R, Yao J, Cai X, Maslov K, et al. Simultaneous functional photoacoustic and ultrasonic endoscopy of internal organs in vivo. Nat Med. 2012;18(8):1297-302.
  4. Zhang HF, Maslov K, Stoica G, Wang LV. Functional photoacoustic microscopy for high-resolution and noninvasive in vivo imaging. Nat Biotechnol. 2006;24(7):848-51.
  5. Zhang HF, Maslov K, Wang LV. In vivo imaging of subcutaneous structures using functional photoacoustic microscopy. Nature protocols. 2007;2(4):797-804.
  6. Brecht HP, Su R, Fronheiser M, Ermilov SA, Conjusteau A, Oraevsky AA. Whole-body three-dimensional optoacoustic tomography system for small animals. Journal of biomedical optics. 2009;14(6):064007.
  7. Longo DL, Stefania R, Aime S, Oraevsky A. Melanin-Based Contrast Agents for Biomedical Optoacoustic Imaging and Theranostic Applications. Int J Mol Sci. 2017;18(8).
  8. Liopo A, Oraevsky A. Nanoparticles as Contrast Agents for Optoacoustic Imaging. 2015. p. 111-49.
  9. Conjusteau A, Liopo A, Tsyboulski D, Ermilov S, Elliott W, Barsalou N, et al. Optoacoustic sensor for nanoparticle linked immunosorbent assay (NanoLISA): SPIE; 2011.
  10. Oraevsky A, Jacques S, Esenaliev R, Tittel F. Laser-based optoacoustic imaging in biological tissues: SPIE; 1994.
  11. Sethuraman S, Amirian JH, Litovsky SH, Smalling RW, Emelianov SY. Spectroscopic intravascular photoacoustic imaging to differentiate atherosclerotic plaques. Opt Express. 2008;16(5):3362-7.
  12. Wang B, Yantsen E, Larson T, Karpiouk AB, Sethuraman S, Su JL, et al. Plasmonic intravascular photoacoustic imaging for detection of macrophages in atherosclerotic plaques. Nano Lett. 2009;9(6):2212-7.
  13. Hannah A, Luke G, Wilson K, Homan K, Emelianov S. Indocyanine green-loaded photoacoustic nanodroplets: dual contrast nanoconstructs for enhanced photoacoustic and ultrasound imaging. ACS Nano. 2014;8(1):250-9.
  14. Wang B, Karpiouk A, Yeager D, Amirian J, Litovsky S, Smalling R, et al. Intravascular photoacoustic imaging of lipid in atherosclerotic plaques in the presence of luminal blood. Opt Lett. 2012;37(7):1244-6.
  15. Wang B, Karpiouk A, Yeager D, Amirian J, Litovsky S, Smalling R, et al. In vivo intravascular ultrasound-guided photoacoustic imaging of lipid in plaques using an animal model of atherosclerosis. Ultrasound Med Biol. 2012;38(12):2098-103.
  16. Zackrisson S, van de Ven S, Gambhir SS. Light in and sound out: emerging translational strategies for photoacoustic imaging. Cancer Res. 2014;74(4):979-1004.
  17. Zhang C, Kimura R, Abou-Elkacem L, Levi J, Xu L, Gambhir SS. A Cystine Knot Peptide Targeting Integrin alphavbeta6 for Photoacoustic and Fluorescence Imaging of Tumors in Living Subjects. J Nucl Med. 2016;57(10):1629-34.
  18. Jokerst JV, Van de Sompel D, Bohndiek SE, Gambhir SS. Cellulose Nanoparticles are a Biodegradable Photoacoustic Contrast Agent for Use in Living Mice. Photoacoustics. 2014;2(3):119-27.
  19. de la Zerda A, Bodapati S, Teed R, May SY, Tabakman SM, Liu Z, et al. Family of enhanced photoacoustic imaging agents for high-sensitivity and multiplexing studies in living mice. ACS Nano. 2012;6(6):4694-701.
  20. Galanzha EI, Zharov VP. In vivo photoacoustic and photothermal cytometry for monitoring multiple blood rheology parameters. Cytometry A. 2011;79(10):746-57.
  21. Zharov VP, Galanzha EI, Shashkov EV, Kim JW, Khlebtsov NG, Tuchin VV. Photoacoustic flow cytometry: principle and application for real-time detection of circulating single nanoparticles, pathogens, and contrast dyes in vivo. Journal of biomedical optics. 2007;12(5):051503.
  22. Cai C, Carey KA, Nedosekin DA, Menyaev YA, Sarimollaoglu M, Galanzha EI, et al. In vivo photoacoustic flow cytometry for early malaria diagnosis. Cytometry A. 2016;89(6):531-42.
